# Effects of Grazing Sheep and Mowing on Grassland Vegetation Community and Soil Microbial Activity under Different Levels of Nitrogen Deposition

Chengyang Zhou [1], Shining Zuo [1], Xiaonan Wang [1], Yixin Ji [1], Qiezhuo Lamao [1], Li Liu [2,*] and Ding Huang [1,*]

1   College of Grassland Science and Technology, China Agricultural University, Beijing 100193, China; s20193040615@cau.edu.cn (C.Z.); s20203243178@cau.edu.cn (S.Z.); b20203240974@cau.edu.cn (X.W.); s20213243257@cau.edu.cn (Y.J.); sy20213243290@cau.edu.cn (Q.L.)
2   Grassland Research Institute, Chinese Academy of Agricultural Sciences, Hohhot 010010, China
*   Correspondence: liuli@caas.cn (L.L.); huangding@cau.edu.cn (D.H.)

**Abstract:** Increasing nitrogen deposition plays a critical role in the material circulation of grassland. Mowing and grazing sheep are important means of utilizing grassland. This study investigated the effects of nitrogen deposition, sheep grazing and mowing on the soil, vegetation and soil microorganisms of grassland. N deposition increased soil inorganic nitrogen, SOM and microbial activity, and decreased soil pH, while grazing sheep and mowing had opposing effects. Compared with mowing, grazing sheep decreased the range of grass groups in the community. N deposition increased the proportion of *Leymus chinensis* in the community and decreased community diversity. N deposition enhanced the contribution rate of soil to the vegetation community, and reduced the effect of microorganisms on the vegetation community. In addition, N deposition significantly interacted with mowing and grazing sheep in terms of effects on soil inorganic nitrogen, soil organic matter (SOM), microbial respiration (Q), microbial mass carbon (MBC), and vegetation diversity. Therefore, appropriate N deposition in sheep grazing and mown grasslands could enhance inorganic N and organic matter, increase microbial activity, offset the adverse effects of grazing sheep and mowing, and contribute to maintaining community diversity and grassland productivity.

**Keywords:** nitrogen deposition; mowing; grazing sheep; microbial activity; grassland diversity

## 1. Introduction

Nitrogen (N) deposition has emerged as a significant anthropogenic factor in global change, with deposition from fossil fuel combustion and artificial fertilizer application [1,2]. As a vital environmental factor limiting the production of terrestrial ecosystems, the rapid increase in N deposition has significantly impacted the productivity and stability of terrestrial ecosystems [3–5]. Low levels of N deposition promote primary production in terrestrial ecosystems and increase ecosystem carbon storage, especially in N-deficient ecosystems [6]. However, it has been suggested that nitrogen enrichment significantly alters the physicochemical environment of the soil. For example. N deposition has been found to reduce the pH value of the soil, leading to soil acidification and negative effects on plants [7,8].

Nitrogen addition eliminates the limitation of this nutrient in the ecosystem, resulting in the rapid growth of grass with high nutrient utilization efficiency, and altered species competition from nutrient to light. The intensity of light competition is a major factor determining the species composition and structure of ecosystems [9,10]. N deposition has been found to alter vegetation composition, including species succession, settlement of dominant species, and loss of rare species, and change the relative abundance of particular functional groups [11,12]. For instance, N deposition significantly increased the aboveground biomass of grass and decreased biomass of legumes [13,14]. Increase in N

deposition results in the loss of species diversity and stability of the community structure in terrestrial ecosystems [2,15–17].

N deposition also profoundly affects the community structure and function of soil microbes. Soil microbial abundance and biomass were found to be affected directly by N deposition and indirectly by changes in soil properties [18,19]. Low-level N addition promoted microbial activity, removed N-limitation by enhancing the net N mineralization rate and N utilization rate, and improved microbial C-limitation by promoting plant growth and litter decomposition [20,21]. However, high levels of N enrichment led to soil acidification, which further created leaching of calcium and magnesium and activated aluminum ions, with toxic effects on microorganisms [22–24].

Both grazing and mowing are indispensable means of grassland utilization, and their effects are multi-faceted. The prevalence of grassland plants is affected by selective grazing by livestock and indiscriminate mowing by humans [25,26]. Grazing also affects topsoil properties [27] and decreases soil aggregate stability and permeability [28–30]. Moderate mowing was found to increase vegetation species richness by removing surface litter and opening lamellar structures [31–35]. Although controversial, proper mowing was reported to reduce energy supplementation to soil microorganisms but to enhance microbial activity [36,37]. The effects of grazing on microorganisms are various, mainly determined by the availability of soil carbon [38,39].

Although the effects of N enrichment on ecosystem structure and function have been studied extensively [1,2,40], our understanding of the effects of N deposition on overall terrestrial ecosystems is still rudimentary, and little is known about how N deposition affects grasslands under grazing and mowing, including their interactions [41–43]. Therefore, we simulated different N deposition levels and grazing and mowing regimes of *Leymus chinensis* grassland, to determine the effects of grazing sheep and mowing on grassland vegetation, soil and microorganisms in the context of background N deposition, to help herdsmen better manage and utilize grassland.

## 2. Materials and Methods

### 2.1. Site Description and Experimental Design

The study was conducted in Hebei, China ($41°45'~41°57'$ N, $115°39'~115°51'$ E, with an MSL of 1400 m), where the average annual temperature is 1.4 °C, the annual accumulated temperature is 1513.1 °C ($\geq$10 °C), the frost-free period is 85~100 days, the annual mean precipitation is between 290 and 400 mm (mainly distributed in July, August, and September), the annual mean wind speed is 4.3 m/s, and the annual sunshine duration is 2930.9 h. In the study area, the soil is dark chestnut soil, with grassland vegetation coverage of between 40–50%, with an average height of approximately 30 cm. *Leymus chinensis* is a constructive species; associated species include *Artemisia frigida*, *Potentillabifurca*, and *Potentilla sericea* [44,45].

A 35 m $\times$ 30 m split-plot was set up on *Leymus chinensis* grassland (Figure 1), with grazing and mowing as the main experimental factors, and N deposition as a secondary factor. The plots included a control plot CK (G0, M0), a light grazing plot (G1), a heavy grazing area (G2) and a mowing area (M1). Consistent with the N deposition pattern in China, it was found that the local natural N deposition amount was about 1.353 g N m$^{-2}$ yr$^{-1}$ [46]. Four sub-plots (3 m $\times$ 5 m) were set up in each plot and urea (N: 46%) was added in the sub-plots (N0: 0 g N m$^{-2}$ yr$^{-1}$, N2: 2.706 g N m$^{-2}$ yr$^{-1}$, N4: 5.412 g N m$^{-2}$ yr$^{-1}$, N8: 10.824 g N m$^{-2}$ yr$^{-1}$). Each combination was repeated three times, with a total of 48 sub-cells (the main area interval was 1 m and the sub-cell interval was 0.5 m) [47].

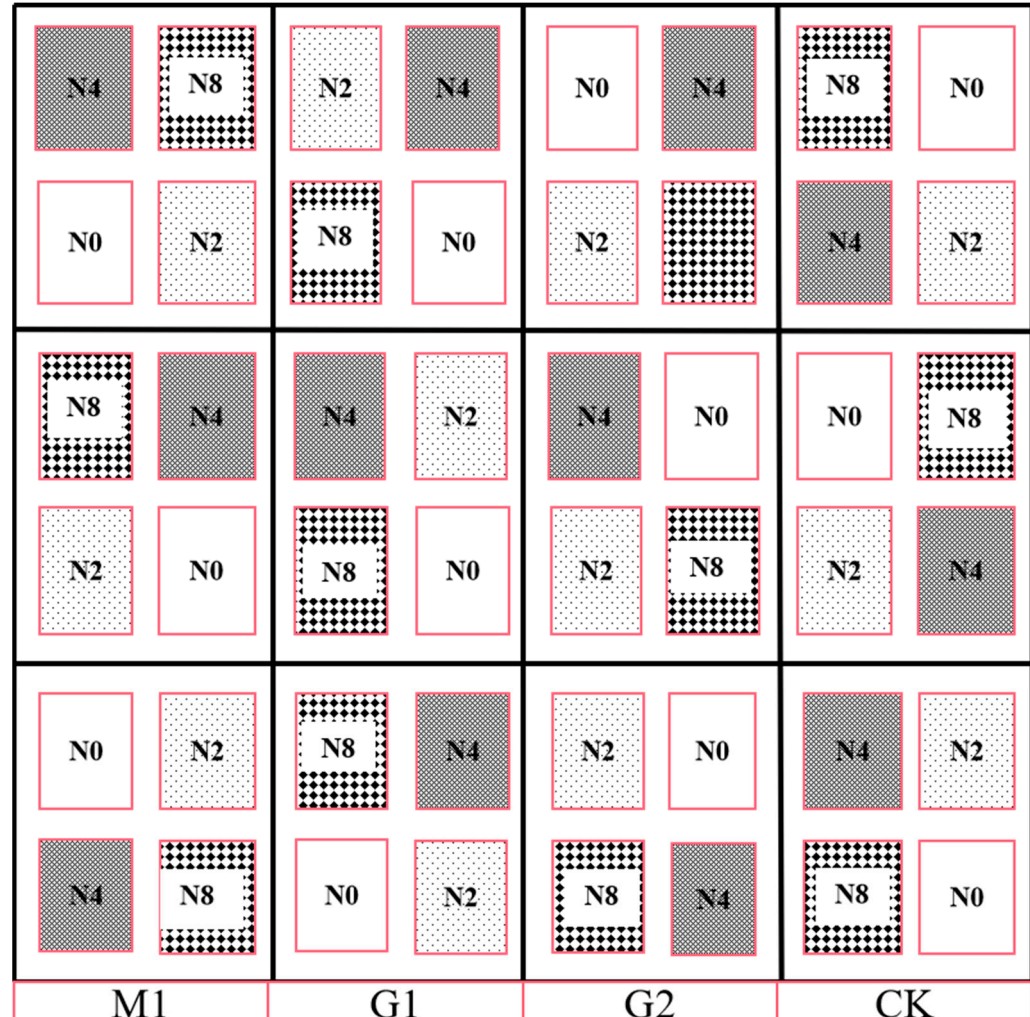

**Figure 1.** The layout of the treatment plots: (M1) Mowing area; (G1) Lightly grazed area; (G2) Heavily grazed areas; (CK) Control area. Different patterns of filling represent nitrogen deposition (N0: 0 g N m$^{-2}$ yr$^{-1}$, N2: 2.706 g N m$^{-2}$ yr$^{-1}$, N4: 5.412 g N m$^{-2}$ yr$^{-1}$, N8: 10.824 g N m$^{-2}$ yr$^{-1}$).

Urea was applied to the grassland before it rained in early May 2015. Grazing sheep and mowing took place in early July of the same year, and the stubble height determined the grazing intensity (G1: 15 cm, G2: 5 cm, M1: 10 cm). Each G1 subplot (15 m$^2$) grazed one adult Mongolian sheep, while the G2 subplot grazed three sheep, all of which were of similar weight. Ropes were placed to measure the height of the forage; when the forage was lowered to the target height, the sheep were driven out of the plot. The grassland was grazed twice a month in July and August, with an interval of 15 days, and mowed at the same time. Before grazing, the sheep were fasted in captivity to minimize the effects of the sheep's feces and urine. Urine deposition by grazing animals, with urea as the major constituent, is an important nitrogen (N) source in grassland ecosystems, and animal feces can improve soil phosphorus, organic matter, and nitrogen (NO$_3$-N and NO$_2$-N) content [48]. Sheep feces and urine could increase the nitrogen deposition in this study and cause errors in the results.

### 2.2. Plant

The above-ground vegetation was sampled during the maturation period in early September and 0.25 m$^2$ sample boxes were randomly placed in the subplot three times without spatial overlap. All live plants in each sample box were classified according to the Inter-Takhta classification system [49]. All plant material on the surface was cut, including

litter and standing dead biomass, and weighed after drying at 65 °C for 48 h. The above-ground biomass yield was estimated using the dry mass of all animate plants in each quadrat, averaged over nine replicates per treatment.

Shannon–Wiener (*H*), ACE (*S*) and Gini indices were used to describe the species composition of the grassland vegetation under different treatments. The Shannon–Wiener (*H*) index describes biodiversity; its value reflects the probability that two individuals belong to two dissimilar species. The ACE (*S*) index is used to estimate the number of species in a community, and is a commonly used index to estimate the total number of species in ecological studies. The mean decrease Gini value compares the importance of variables by calculating the effect of each variable on the heterogeneity of observations at each node of a classification tree using the Gini index—the greater the value, the greater the importance of the variable.

Shannon–Wiener (*H*) [50], ACE (*S*) [51] and Gini [52] indices were calculated according to the following formula:

$$H_{shannon} = -\sum p_i \ln p_i \tag{1}$$

$$S_{ACE} = \begin{cases} S_{abund} + \dfrac{S_{rare}}{C_{ACE}} + \dfrac{n_1}{C_{ACE}} \hat{\gamma}^2{}_{ACE}, \ for \ \hat{\gamma}_{ACE} < 0.80 \\ S_{abund} + \dfrac{S_{rare}}{C_{ACE}} + \dfrac{n_1}{C_{ACE}} \widetilde{\gamma}^2{}_{ACE}, \ for \ \widetilde{\gamma}_{ACE} \geq 0.80 \end{cases} \tag{2}$$

$$\text{Gini} = 1 - \sum p_i{}^2 \tag{3}$$

where,

$N_{rare} = \sum_{i=1}^{abund} i n_i, C_{ACE} = 1 - \dfrac{n_1}{N_{rare}},$

$\widetilde{\gamma}^2{}_{ACE} = max\left[ \widetilde{\gamma}^2{}_{ACE} \left\{ 1 + \dfrac{n_{rare}(1-c_{rare})\sum_{i=1}^{abund} i(i-1)n_i}{n_{rare}(n_{rare}-c_{rare})} \right\}, 0 \right],$

$\hat{\gamma}^2{}_{ACE} = max\left[ \dfrac{S_{rare}}{C_{rare}} \dfrac{\sum_{i=1}^{abund} i(i-1)n_i}{n_{rare}(n_{rare}-1)} - 1, 0 \right],$

$p_i$ = n/N,

*n* = the number of individuals in species,

*N* = the total number of individuals in the area,

*ln* = the natural log,

$n_i$ = the number of species with i individuals,

$S_{rare}$ = the number of species with 'abund' or fewer individuals,

$S_{abund}$ = the number of species with more than 'abund' individuals,

*abund* = the threshold to be considered an 'abundant' species; this is set to 10 by default.

### 2.3. Soil Properties and Microorganisms

After the plants were cut, three soil cores were randomly drilled with a depth of 0–15 cm and a diameter of 3.5 cm for the soil drills in each plot box. Each subplot sample contained nine soil cores, without plant roots and stones, which were mixed evenly and immediately stored at 4 °C.

The pH was determined with a pH meter (FE20-FiveEasy™, Mettler Toledo, Zurich, Switzerland) in a soil: water (1:2.5) solution (*w/v*). An amount of 10 g of the air-dried soil sample was weighed after passing through a 2 mm sieve and 25 mL of deionized water was added. The mixture was shaken for 30 min and let stand for 30 min. The pH value of the suspension was measured using an acidity meter. Nitrate-N ($NO_3$-N) and ammonium-N ($NH_4$-N) was extracted with 2 M KCl. $NH_4$-N was measured using the salicylate method, while $NO_3$-N was measured using the cadmium reduction method on a FIAstar 5000 Analyzer (FIAstar 5000 Analyzer; Foss Tecator, Hillerød, Denmark). Soil total N was measured using the FOSS Kieltec apparatus, and soil organic matter was measured by potassium dichromate dilution calorimetry.

Soil microbial biomass carbon (MBC) was estimated using a chloroform fumigation extraction method [53]. Microbial nitrogen (MBN) was determined by the chloroform fumigation extraction–ninhydrin colorimetric method [54]. A small beaker of fresh soil containing 12.5 g passed through a 2 mm sieve was placed into a vacuum dryer with

three beakers containing chloroform, water and dilute NaOH. The chloroform was then vacuumed to boil vigorously for 3–5 min and left in a dark room for 24 h. After fumigation and all the chloroform was gone, the same weight of soil was placed in another dryer and the process repeated without the chloroform. All the fumigated soil samples were transferred to 150 mL triangular bottles, 50 mL 0.5 mol/L $K_2SO_4$ (soil-water ratio: 1:4) was added, and the extraction was filtered after 30 min of shaking. The unfumigated soil samples were subjected to the same operation, and a blank was made at the same time. The organic carbon content of the extract was determined using a Shimadzu TOC500 organic carbon analyzer to calculate the MBC. A quantity of 10 ml of soil extract was placed in a 25 mL calibration tube with a plug, and the same volume of ninhydrin solution was added. After being fully mixed, the solution was added to a pressure cooker for oxidation for at least 0.5 h. The solution was removed and cooled to room temperature, and colorimetry was performed at 275 nm to calculate the MBN. Soil organic matter was determined by potassium dichromate dilution calorimetry. An amount of 0.5000 g of air-dried soil sample was weighed and passed through a 0.149 mm sieve. An amount of 10 mL of 1 mol/L $(1/6\ K_2Cr_2O_7)$ was added to the solution, mixed and 20 mL of concentrated $H_2SO_4$ was added. The solution was cooled for 30 min, diluted to 250 mL with water, and three drops of phenanthroline indicator were added. The solution was then titrated with 0.5 mol/L $FeSO_4$ standard solution (color change from green to dark green to brick red). Three blank tests were performed with the same weight of silica instead of soil samples.

Soil microbial respiration was measured by the lye absorption method [55]. Fresh soil equivalent to 20 g of dry soil was put into a 500 mL glass flask, and 20 mL 0.1 mol/L NaOH solution was poured into the flask to absorb $CO_2$ exhaled from the soil. The soil sample was wrapped with a double layer of gauze and suspended in the glass bottle, and the bottle was plugged tightly. The bottle was placed in an incubator at 25 °C for 7 days after dark culture, and the remaining NaOH was titrated with 0.1 mol/L HCL. Microbial respiration entropy $(qCO_2)$ was determined according to the method of Wardle and Ghani [56].

### 2.4. Statistical Analyses

Analysis of variance (one-way ANOVA, SPSS 26.0) followed by Duncan's post hoc test was used to test the effects of different treatments or N deposition on total plant biomass, Shannon's (*H*) index, the abundance coverage-based estimator (ACE), soil properties, and microbial parameters. The interaction of N deposition and grazing and cutting was tested by GLM bivariate analysis (two-way ANOVA, SPSS26.0)

Redundancy analysis (RDA) was utilized to understand the relationship between vegetation functional groups, soil properties, and microbial characteristics under different modes of grassland utilization (with N deposition). Random forest modeling was used to forecast the survival of plant functional groups (family level) in grasslands with N deposition under different uses. The redundancy analysis (RDA) and random forest model were performed on the BMK Cloud Platform (http://www.biocloud.net, accessed on 1 June 2022), an online analysis website. Variance partitioning analysis (VPA) was used to measure the impact of soil factors and microbial properties on changes in vegetation communities in the context of N deposition. The soil properties and microbial properties were measured as two types of environmental factors, and the explanatory rates of these two types of environmental factors on changes in plant community structure were analyzed. The *varpart* function in R was used for VPA analysis to obtain the explanatory rates of variance of each part, and then the *showvarpart* function in the *vegan* package in R was used to draw a Venn diagram.

## 3. Results

### 3.1. Responses of Soil and Microbes to Grazing Sheep and Mowing under Nitrogen Deposition Conditions

In all treatments, TN, $NO_3$-N, and $NH_4$-N increased significantly across the N deposition gradient (Table 1). The pH was significantly lower in both the heavy grazing area (G2)

and the control area (CK) under the highest N deposition (N8) but was not affected by N deposition under the moderate grazing (G1) and mowing treatments (M). N deposition significantly increased the soil organic matter (SOM) content in the grazing plots (G) and control plots (CK) but did not affect SOM in the mowing plots (Table 1).

**Table 1.** Soil properties of grazing sheep and mowing grasslands in the context of different levels of nitrogen deposition.

| Soil Properties | Treatment | Nitrogen Deposition (g N m$^{-2}$ yr$^{-1}$) | | | | ANOVA | | |
|---|---|---|---|---|---|---|---|---|
| | | 0 | 2.706 | 5.412 | 10.824 | N | T | N * T |
| pH | CK | 7.76 ± 0.02 [aC] | 7.80 ± 0.02 [aC] | 7.74 ± 0.03 [aB] | 7.59 ± 0.02 [bB] | | | |
| | G1 | 7.84 ± 0.04 [aBC] | 7.82 ± 0.04 [aBC] | 7.76 ± 0.11 [aAB] | 7.74 ± 0.06 [aA] | *** | *** | NS |
| | G2 | 8.00 ± 0.01 [aA] | 8.00 ± 0.03 [aA] | 7.78 ± 0.07 [bAB] | 7.76 ± 0.02 [bA] | | | |
| | M1 | 7.89 ± 0.07 [aAB] | 7.93 ± 0.04 [aAB] | 7.89 ± 0.02 [aA] | 7.85 ± 0.05 [aA] | * | *** | NS |
| TN, g/kg | CK | 3.30 ± 0.00 [cAB] | 3.48 ± 0.03 [bA] | 3.73 ± 0.03 [aA] | 3.67 ± 0.04 [aA] | | | |
| | G1 | 3.02 ± 0.09 [cC] | 3.17 ± 0.15 [bcC] | 3.36 ± 0.08 [abB] | 3.55 ± 0.07 [aB] | *** | *** | NS |
| | G2 | 3.13 ± 0.01 [cBC] | 3.31 ± 0.04 [bB] | 3.48 ± 0.04 [aB] | 3.59 ± 0.05 [aB] | | | |
| | M1 | 3.38 ± 0.04 [bA] | 3.42 ± 0.02 [bA] | 3.42 ± 0.05 [bB] | 3.52 ± 0.02 [aB] | *** | ** | ** |
| NO$_3$-N, g/kg | CK | 0.41 ± 0.09 [cC] | 0.64 ± 0.06 [cC] | 6.80 ± 0.13 [bC] | 16.77 ± 0.55 [aB] | | | |
| | G1 | 0.49 ± 0.08 [dC] | 3.52 ± 0.23 [cB] | 9.60 ± 0.11 [bB] | 20.52 ± 0.66 [aA] | *** | *** | *** |
| | G2 | 1.70 ± 0.03 [cA] | 4.25 ± 0.59 [cA] | 17.36 ± 0.04 [bA] | 21.82 ± 1.00 [aA] | | | |
| | M1 | 0.77 ± 0.05 [dB] | 4.72 ± 0.47 [cAB] | 6.73 ± 0.03 [bC] | 17.03 ± 0.25 [aB] | *** | *** | *** |
| NH$_4$-N, g/kg | CK | 1.39 ± 0.08 [dA] | 1.71 ± 0.14 [cA] | 1.98 ± 0.05 [bA] | 2.53 ± 0.06 [aA] | | | |
| | G1 | 1.41 ± 0.06 [cA] | 1.57 ± 0.06 [bcA] | 1.71 ± 0.10 [abB] | 1.89 ± 0.02 [aB] | *** | *** | ** |
| | G2 | 1.04 ± 0.11 [cB] | 1.33 ± 0.05 [bB] | 1.67 ± 0.11 [aB] | 1.63 ± 0.02 [aC] | | | |
| | M1 | 1.23 ± 0.50 [cAB] | 1.69 ± 0.04 [bA] | 1.87 ± 0.06 [abAB] | 2.00 ± 0.08 [aB] | *** | ** | * |
| SOM, g/kg | CK | 42.76 ± 0.60 [cA] | 46.07 ± 0.13 [bA] | 46.62 ± 0.86 [bA] | 49.23 ± 0.71 [aA] | | | |
| | G1 | 36.81 ± 0.18 [bB] | 38.22 ± 0.65 [bC] | 38.44 ± 0.86 [bC] | 44.88 ± 0.45 [aC] | *** | *** | *** |
| | G2 | 32.43 ± 1.04 [cC] | 39.96 ± 0.93 [bC] | 43.14 ± 0.14 [bB] | 47.34 ± 0.59 [aB] | | | |
| | M1 | 42.01 ± 0.45 [aA] | 42.92 ± 0.30 [aB] | 43.19 ± 0.24 [aB] | 42.12 ± 0.38 [aD] | *** | ** | *** |

Different lowercase letters in the same row indicate significant differences among treatments with different nitrogen deposition levels under the same grassland utilization mode ($p < 0.05$); different capital letters in the same column indicate significant differences among treatments with the same nitrogen deposition level and different grassland utilization modes ($p < 0.05$). Significant effects of treatments are marked with an asterisk. Single, double, and triple stars indicate significance levels at $p < 0.5$, $<0.01$ and $<0.001$, respectively, and NS indicates no significant difference.

Both grazing sheep and mowing significantly decreased soil NH$_4$-N and TN in grasslands with high N input. Grazing sheep (G) significantly increased soil NO$_3$-N at all N deposition levels (N0, N2, N4, N8) in this study, while mowing only had a positive effect on soil NO$_3$-N at low N deposition levels (N0, N2). No significant difference in soil microbial respiration (MR) was found between the heavily grazing area (G2) and the control area (CK) at each N deposition level (N0→N8), while, in the moderate grazing areas (G1) and mowing areas (M1), the highest levels of N addition (N8) significantly increased the MR. Both grazing and mowing significantly reduced soil organic matter (SOM) with N addition (Table 1).

The microbial biomass carbon (MBC) content of all treatments increased gradually across the N deposition gradient and was significantly different from the control treatment (CK) at higher levels of N addition (N4, N8) (Figure 2a). Higher levels of N deposition (N4, N8) had no significant effect on microbial biomass N (MBN) in the moderate grazing

area (G1) but caused a significant increase in MBN in other areas (CK, G2, M) (Figure 2b). Whether or not N was added to the grassland, grazing (G) and mowing (M) significantly decreased microbial carbon and nitrogen (MBC, MBN). Under the same N deposition level, microbial carbon and nitrogen (MBC, MBN) contents were ranked in order from high to low: CK > M1 > G1 > G2 (Figure 2).

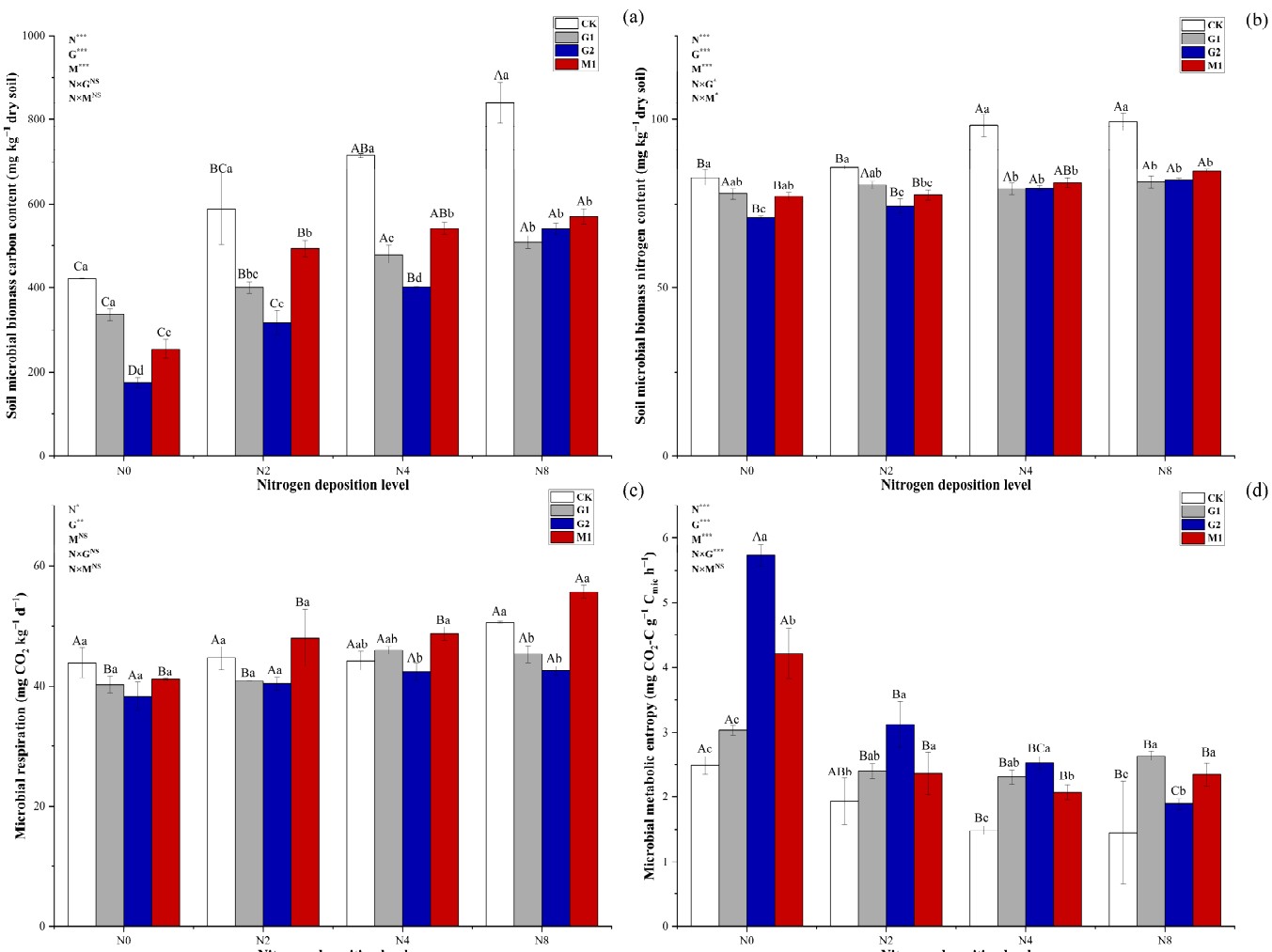

**Figure 2.** Effects of grazing sheep (G1, G2) and mowing (M1) on soil microorganisms at different nitrogen deposition levels (N0, N2, N4, N8); (**a**) soil microbial biomass carbon; (**b**) soil microbial biomass nitrogen; (**c**) microbial respiration; (**d**) microbial metabolic entropy. Different lowercase letters indicate significant differences among different treatments under the same nitrogen deposition level ($p < 0.05$); different capital letters indicate significant differences among different nitrogen deposition levels under the same treatment ($p < 0.05$). Significant effects of treatments are marked with an asterisk. Single, double, and triple asterisks indicate the significance levels at $p < 0.5$, $<0.01$, and $<0.001$, respectively, and NS indicates no significant difference.

No significant difference in soil microbial respiration (MR) was found between the heavy grazing area (G2) and the control area (CK) at each N deposition level (N0→N8), while in moderate grazing areas (G1) and mowing areas (M1), the highest levels of N addition (N8) significantly increased MR (Figure 2c). At lower N addition levels (N2), microbial respiration (MR) did not differ significantly among all treatments in this study. In contrast, when N enrichment reached the highest value (N8), either light grazing (G1) or heavy grazing (G2) significantly reduced the MR (Figure 2c). Soil microbial metabolic entropy (Q) decreased significantly with N deposition, whereas grazing and mowing

increased it significantly. N deposition showed a significant interaction with grazing sheep but not mowing (Figure 2d).

### 3.2. Responses of Plants to Grazing Sheep and Mowing under Nitrogen Deposition Conditions

At high N deposition levels (N4, N8), the importance values of *Leymus chinensis* decreased significantly on the grazing gradients but showed no statistical regularity on the N deposition gradient (Figure 3a). Plant aboveground biomass was decreased with grazing gradient in the presence of N input in the grassland but increased with the N deposition gradient in the control plot (CK), reaching a maximum at N8 (Figure 3b).

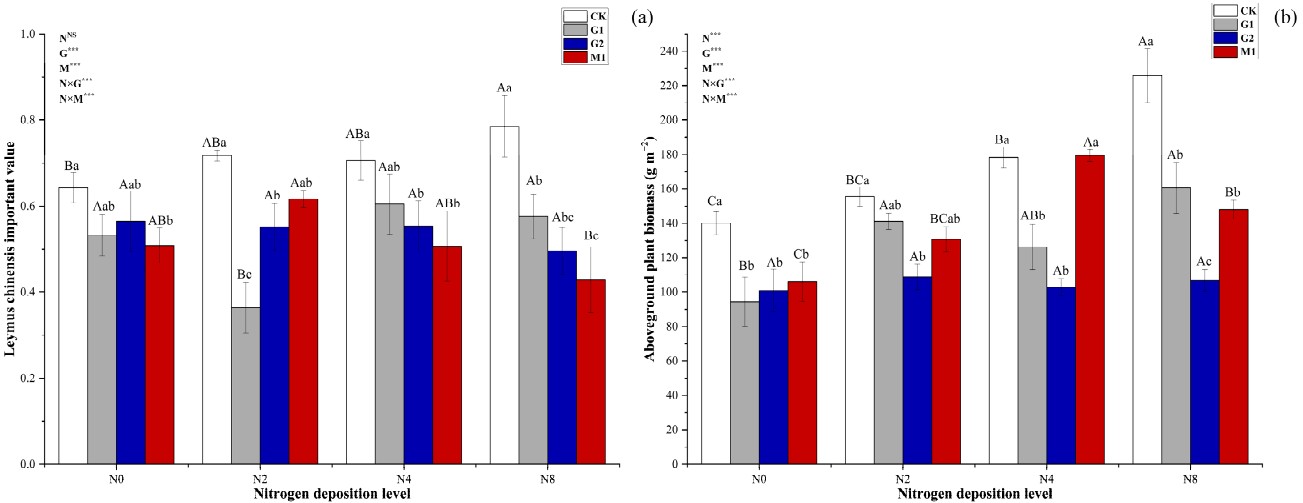

**Figure 3.** Effects of grazing sheep (G1, G2) and mowing (M1) on plants at different nitrogen deposition levels (N0, N2, N4, N8); (**a**) *Leymus chinensis* importance value; (**b**) Above-ground plant biomass. Different lowercase letters indicate significant differences among different treatments under the same nitrogen deposition level ($p < 0.05$); different capital letters indicate significant differences among different nitrogen deposition levels under the same treatment ($p < 0.05$). Significant effects of treatments are marked with an asterisk. Triple asterisks evaluate the significance level at $p < 0.001$, and NS indicates no significant difference.

It was found that both grazing (G) and mowing (M) significantly increased the Shannon (*H*) index (Figure 4a). Low levels of N deposition (N2) did not affect the grassland community of the Shannon (*H*) index. On the contrary, higher levels of N input (N4, N8) significantly decreased the Shannon (*H*) index in grazing areas (G) (Figure 4a).

It was found that the ACE (*S*) of the control plot (CK) decreased with the N deposition gradient (Figure 4b). N deposition significantly increased the ACE in the middle grazing area (G1) and, conversely, significantly decreased the ACE (*S*) in the heavy grazing area (G2) (Figure 4b). In the absence of N (N0), light grazing (G1) and mowing (M) significantly decreased the ACE (*S*), while heavy grazing (G2) significantly increased the ACE (*S*) (Figure 4b).

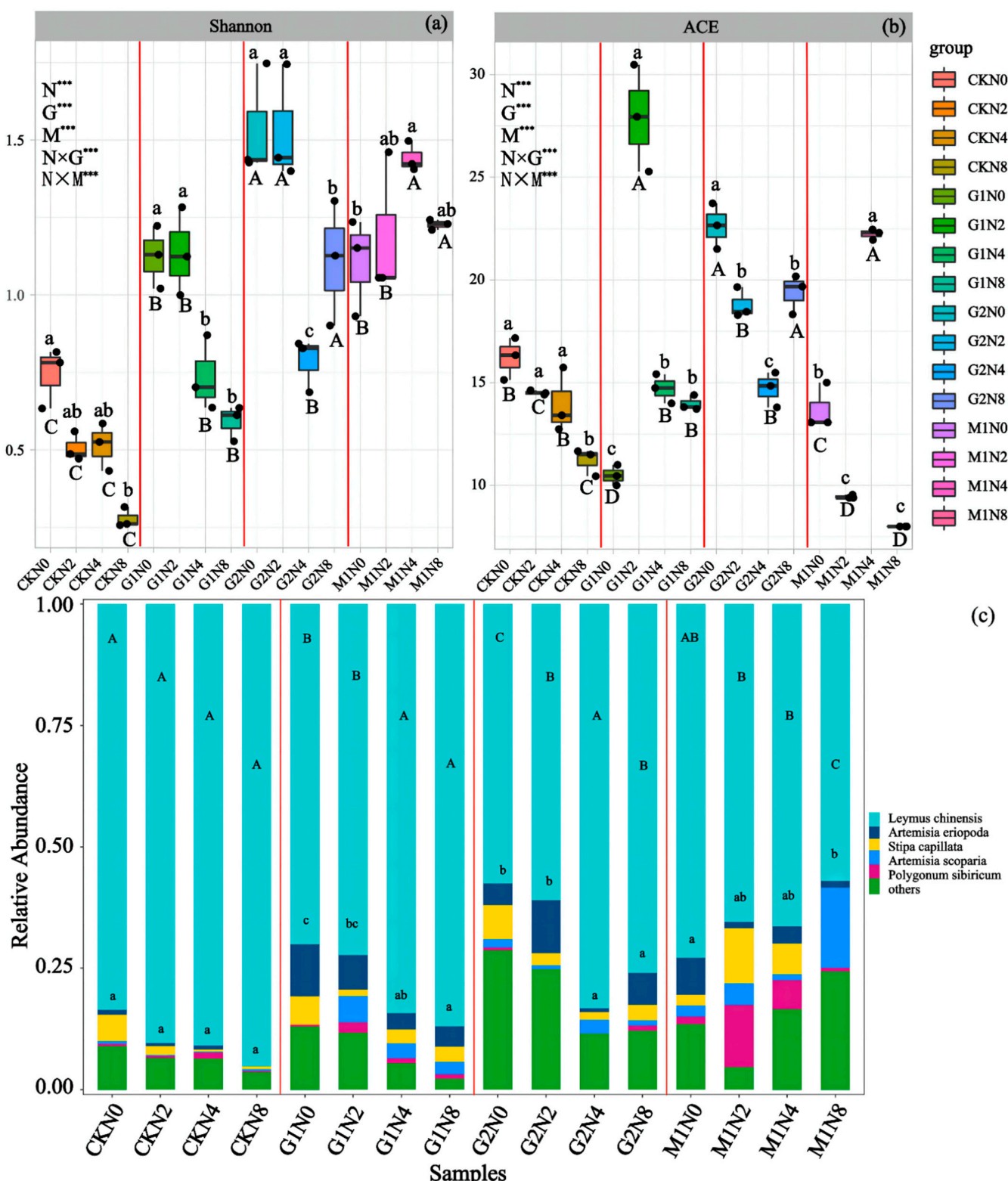

**Figure 4.** Effects of grazing sheep (G1, G2) and mowing (M1) on grassland plant community at different nitrogen deposition levels (N0, N2, N4, N8); (**a**) Shannon index; (**b**) Abundance coverage-based estimator (ACE); (**c**) Relative abundance of species. Different lowercase letters indicate significant differences among different treatments under the same nitrogen deposition level ($p < 0.05$); different capital letters indicate significant differences among different nitrogen deposition levels under the same treatment ($p < 0.05$). Significant effects of treatments are marked with an asterisk. Triple asterisks indicate the significance level at $p < 0.001$.

The study area's top five relative abundance species were *Leymus chinensis, Artemisia eriopoda*, *Stipa capillata*, *Artemisia scoparia*, and *Polygonum sibiricum* (Figure 4c). The relative abundance of *Leymus chinensis* increased on the N deposition gradient in the control plot (CK) and conversely decreased with the N deposition gradient in the mowing plot (M) (Figure 4c). Low levels of N addition (N0, N2) had no significant effect on the relative abundance of *Leymus chinensis*, while high levels of N deposition (N4, N8) significantly increased the relative abundance of *Leymus chinensis* in the grazing area (G) (Figure 4c).

Grazing sheep and mowing significantly reduced the relative abundance of *Leymus chinensis* at N2 levels (Figure 4c). When N addition reached the highest level (N8), both heavy grazing (G2) and mowing (M) significantly reduced the relative abundance of *Leymus chinensis* compared with moderate grazing or no grazing (G1 or CK) (Figure 4c).

With increase in grazing intensity, the relative abundance of *Gramineae, Leguminosae*, and *Cyperaceae* decreased significantly, while the relative abundance of *Rosaceae, Umbelliferae, Compositae*, and *Lioaceae* increased significantly (Figure 5c). It was found that, in the grazing area, according to the mean decrease Gini value, the plants' classification ranked from large to small was (*Rosaceae > Asteraceae > Apiaceae > Poaceae > Polygonaceae > Iridaceae > Fabaceae*) (Figure 5a). Mowing led to a significant decrease in the vegetation of *Gramineae* and *Cyperaceae* and a significant increase in *Compositae, Umbelliferae, Liaoaceae*, and *Leguminosae* (Figure 5c). The mean decrease Gini value of *Poaceae* in the grazing areas was lower than that of some species, while the mean decrease Gini value of *Poaceae* was the highest in the mowing areas (Figure 5a,b).

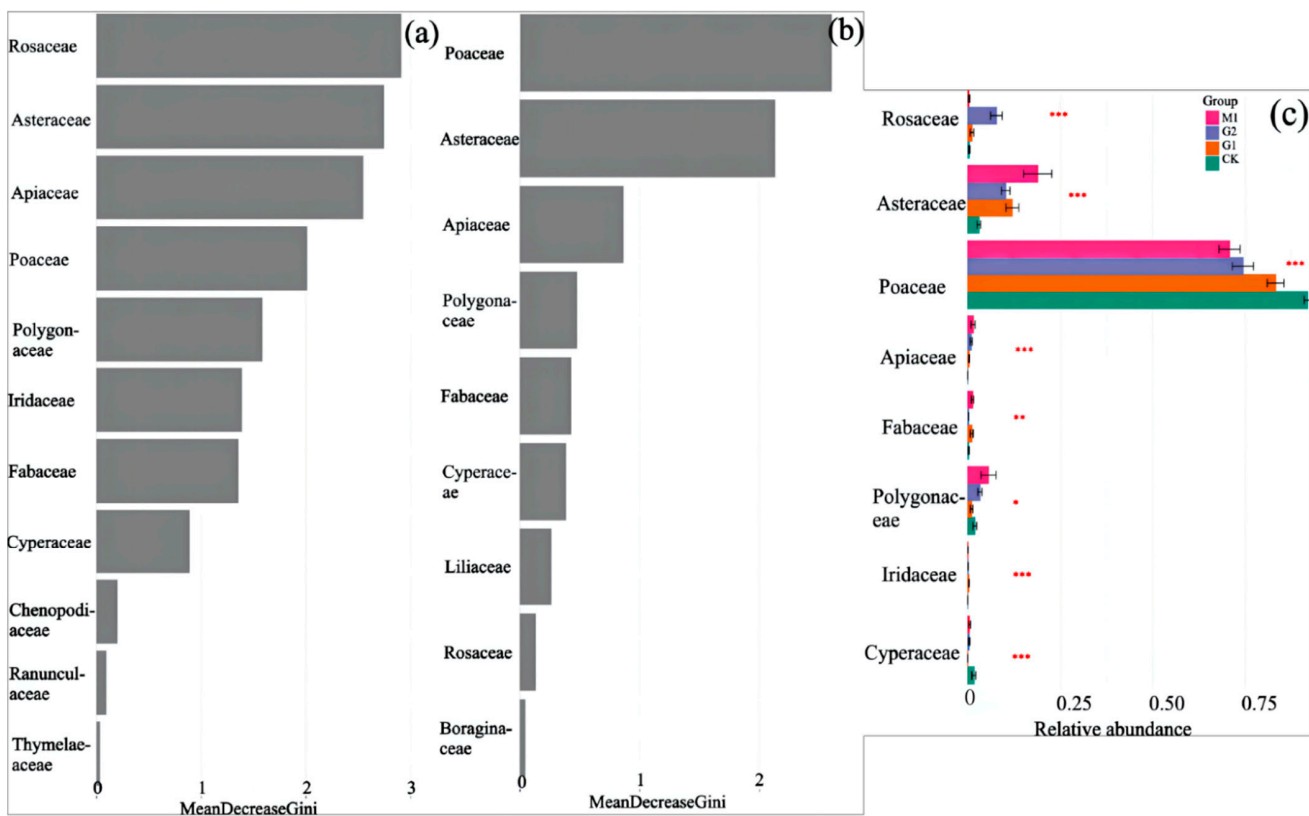

**Figure 5.** Predicting the pattern of grazing sheep and mowing grassland vegetation functional groups (family classification level) in context of nitrogen deposition based on a random forest model; (**a**) Mean decrease Gini values of plant functional groups in grazing sheep grasslands in the context of nitrogen deposition; (**b**) Mean decrease Gini values of plant functional groups in mowing grasslands in the context of nitrogen deposition; (**c**) Relative abundance of functional groups. Significant effects of treatments are marked with an asterisk. Single, double, and triple asterisks evaluate the significance levels at $p < 0.5$, <0.01 and <0.001, respectively.

### 3.3. Responses of the Soil-Plant-Microbe System to Grazing Sheep and Mowing under Nitrogen Deposition Conditions

According to RDA analyses, it was found that the soil properties and the microbial biomass explained more than 90% of the vegetation composition. Plant composition was significantly positively correlated with soil properties ($NO_3$-N, $NH_4$-N and TN) at high levels of N addition (N8) but negatively or not correlated with the pH (Figure 5). In contrast, plant composition was positively correlated with the soil pH at low levels of N addition (N2) but negatively or not correlated with other soil factors (Figure 6). The results showed that there was a significant negative correlation between soil pH and other soil properties, while there was a significant positive correlation of soil properties (SOM, TN, $NO_3$-N, $NH_4$-N) in the CK and G2 plots (Figure 6).

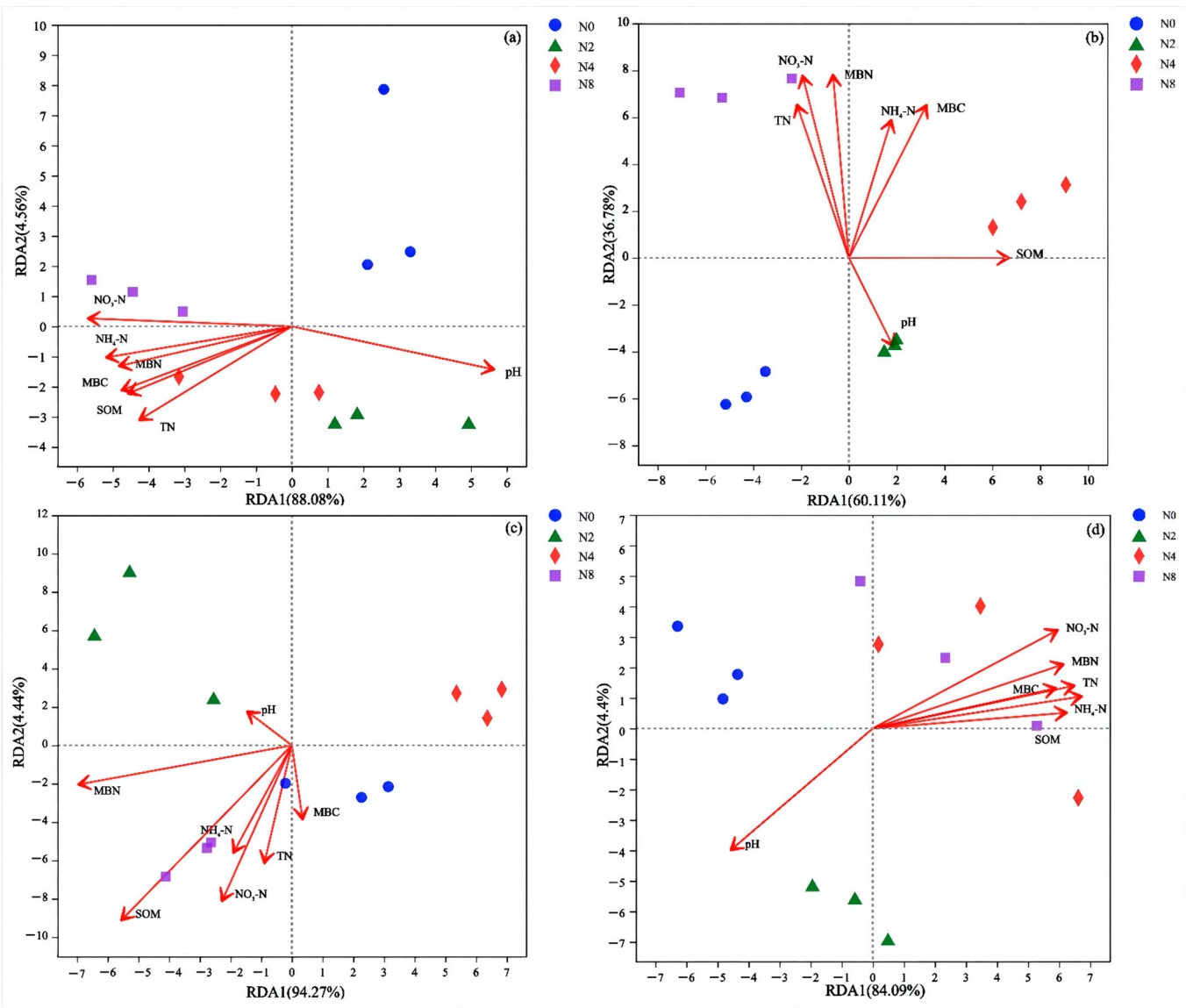

**Figure 6.** Redundancy analysis (RDA) results of soil and microbial properties and vegetation functional groups in grazing sheep and mowing grasslands under different nitrogen deposition backgrounds: (**a**) CK; (**b**) M1; (**c**) G1; (**d**) G2. Significance ($p < 0.05$) was assessed using 999 permutations of the complete RDA. pH: soil potential of hydrogen; TN: soil total nitrogen; $NO_3$-N: soil nitrate-nitrogen; $NH_4$-N: soil ammonium nitrogen; SOM: soil organic matter; MBC: soil microbial biomass nitrogen; MBN: soil microbial biomass carbon.

RDA analyses did not quantify the impact of soil and microbial systems on vegetation functional groups. Therefore, we performed a further analysis of variance decomposition (VPA); the results showed that soil properties and microbial biomass (MBC, MBN) explained 94.43% of vegetation community variation under N deposition (N8) (Figure 7). Compared with N0 (Figure 7a), the high level of N input (N8) reduced the individual interpretability of microorganisms to vegetation communities by 7.69% (Figure 7b), and the joint interpretability of soil and microorganisms to vegetation decreased by 17.79%. In contrast, the individual interpretability of soil to vegetation communities increased by 30.73% (Figure 7).

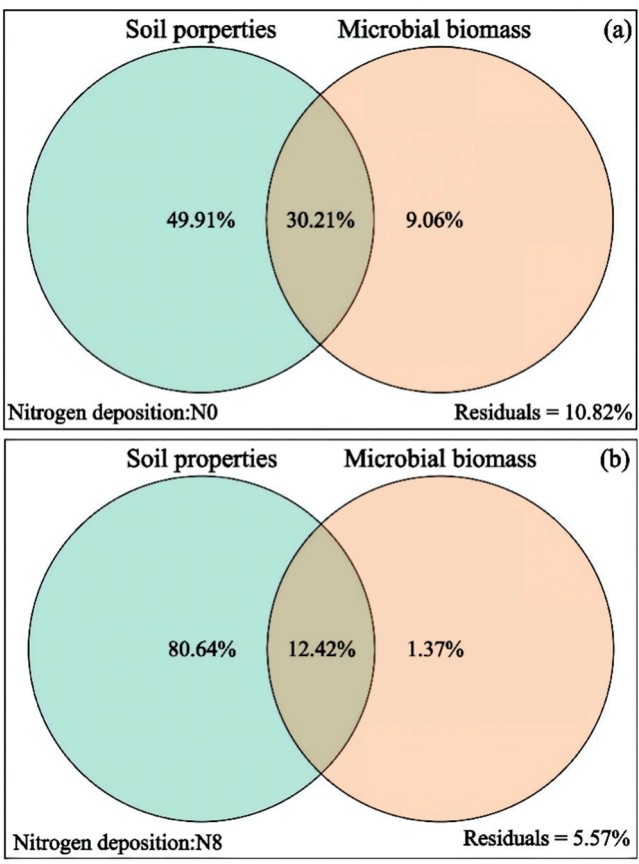

**Figure 7.** Variation partitioning analysis of the relative contributions of soil properties and microbial biomass to variation in plant diversity; (**a**) Nitrogen deposition level is N0; (**b**) Nitrogen deposition level is N8.

## 4. Discussion

It was found that simulated N deposition significantly decreased soil pH and significantly increased TN, inorganic N ($NO_3$-N, $NH_4$-N), and SOM, these results confirming the observations of Liu et al. [20,57] on N deposition. Previous studies demonstrated that grazing mainly affects the physicochemical properties of the topsoil [27], and over-grazing reduces soil aggregate stability and permeability [28–30]. Similar to the findings in this study, a meta-analysis based on years of grazing experiments showed that grazing slightly increased soil pH in northern China grasslands, and pH increased with grazing gradients [58]. An observed effect, also described by Throop, was that continued input of N led to excessive accumulation of ammonium N, which triggered nitrification, producing an increase in $H^+$, and causing soil acidification [59]. This study found that N deposition significantly increased TN, $NH_4$-N, and $NO_3$-N in grazing grassland soils; previous studies also found that N addition in both grazing and non-grazing plots increased inorganic N content in the soil [60]. Grazing and mowing reduced TN content, which has been reported

in some grassland literature, confirming part of the results in this study, except that TN did not change significantly in the mowing area [61]. A loss of soil organic matter due to atmospheric N deposition was recorded in a previous study [62]. However, it was found that N deposition significantly increased SOM in grazing and control plots but did not affect the SOM content in cutting areas. A possible reason is that gramineous plants (*Leymus chinensis*) have a strong ability to use N. In addition, low levels of N input did not cause excessive N, and a small amount of N combined with organic matter and SOM content did not show any substantial reduction [59].

There are various opinions concerning the effects of grazing on soil organic matter. Frank et al. [63] considered that organic matter content in heavy grazing areas did not decrease as C4 plants with strong organic matter production capacity markedly increased. However, many studies have found that grazing and cutting drastically reduced the SOM by reducing the above-ground biomass of plants and the return of the litter to the soil, which is consistent with the results obtained here [63,64].

Numerous studies have shown that microbial biomass carbon and nitrogen (MBC, MBN), and microbial respiration (MR), increase with a lower level of N deposition gradient, which was also confirmed in this study [65–69]. In addition, it was noted that MBC increased with the N gradient in grazing and mowing grassland, but the MBN was relatively stable. Some researchers have noted that when N enrichment exceeds a specific limit, the soil pH decreases. Soil acidification limits microorganisms as a result of decrease in $Mg^{2+}$ and an increase in $Al^{3+}$ causing toxicity [20–24].

Contrary results have been reported, where microbial biomass C and N increased significantly in grazing grasslands but tended to decrease with increasing grazing gradients [70]. However, it was found that both grazing and mowing resulted in a significant decrease in microbial carbon and nitrogen (MBC, MBN), no matter how great the N deposition. Possible reasons suggested for this included differences in livestock species and numbers and lack of manure return.

Previous reports on N deposition have often been controversial. Some researchers held that soil microbial respiration was inhibited by N deposition in the early stage [71,72], while others held different views that short-term simulated N deposition significantly promoted soil microbial respiration [73]. Interestingly, this study found that N deposition did not dramatically affect microbial respiration in grazing or mowing grassland. It was demonstrated that microbial metabolic entropy (Q), representing the ratio of mineralizable carbon to microbial biomass carbon, could sensitively reflect the impact of environmental factors and changes in management measures on microbial activity [74,75]. A previous study indicated that N deposition did not affect microbial metabolic entropy but this increased considerably when N deposition and mowing interacted [76]. Conversely, another study demonstrated that soil microbial metabolic entropy (Q) increased with progressively stressful grazing, which was consistent with our findings [70].

In addition, it was found that N deposition helped to suppress the increase in Q produced by grazing and mowing, suggesting that N fertilization to grazing grasslands was more beneficial to soil carbon sequestration than no N application. Moreover, it was found that grazing rather than mowing interacted with N deposition [77,78].

Although statistically insignificant, with a background of N deposition, this study indicated that the constructive species *Leymus chinensis* gradually increased with the gradient of N deposition without disturbance but was relatively stable during grazing sheep and cutting. It was found that N enrichment significantly increased the vegetation biomass of the light grazing plot (G1) and control plot (CK) but did not affect the heavy grazing plot (G2). Surprisingly, as also demonstrated by Zong et al. [60], the biomass of the cutting area (M1) with N4 level N addition was significantly higher than that without any treatment.

With respect to the ACE (*S*) and the Shannon (*H*) indices for the grazing plot and cutting plot under different N deposition levels, it was found that higher N deposition (N4, N8) reduced the Shannon (*H*) and ACE (*S*) index values regardless of interference,

indicating loss of community diversity. Similar to this study, many authors have suggested that increase in N deposition leads to decline in species diversity in the ecosystem [2,15,79].

It is well-established that grazing plays a vital role in generating plant diversity owing to the selective feeding of livestock [80]. Todd et al. [26] emphasized that grazing led to changes in community structure by changing the relative abundance of different functional groups. For example, grazing sheep inhibited tall, erect plants and tufted plants [81].

In addition, it was found that when grazing sheep on *Leymus chinensis* grassland with N input, the plant species diversity value, Shannon index (*H*), and the ACE (*S*) index did not decrease but, instead, increased. For example, grazing sheep increased the abundance of *Rosaceae, Asteraceae, Apiaceae* and *Polygonaceae*. Some early studies showed that grazing sheep disturbance helped to maintain biodiversity because the competitiveness of dominant species was inhibited, which provided conditions for the settlement of other species, and led to an increase in species diversity [6,21,82,83]. It was observed that the species diversity (Shannon index) of the mowing plots (M1) with high N enrichment was relatively stable. Nevertheless, ACE (*S*) decreased significantly, indicating that some shorter plants (Figure 5c) accelerated their growth in a short period after the dominant species was cut, which increased biodiversity.

Similar to this study, many studies on community diversity have shown that, although cutting inhibited the competitiveness of dominant species in a short time, *Gramineae* plants with high nutrient utilization efficiency grew rapidly in fertilized grassland and still maintained a leading position in light competition [84]. The pattern of plant competition for resources was not changed [9,10]. Therefore, mowing plots would eventually be covered with *Leymus chinensis*, resulting in the loss of other species.

The spatial coexistence of grassland vegetation functional groups (family level) was predicted via random forest modeling [85]. This confirmed that the Gini value of the *Gramineae* functional group in the grazing area ranked lower than that in the cutting area, and that the impact of other functional groups (such as *Rosaceae*, *Asteraceae*, and *Apiaceae*) on plant community structure increased. However, the *Gramineae* was still the main functional group in the cutting area, and the coexistence pattern of vegetation was not affected.

For soil-plant-microbe system synthesis, the RDA analysis showed that high-level N deposition (N8) strengthened the positive correlation between vegetation communities and soil properties (except pH) and changed the pattern of vegetation communities affected by soil properties. In contrast, low-level N deposition did not change the correlation between vegetation communities and soil properties, which was confirmed in a paper by Cleland and Sinsabaugh [11,86].

Variance partitioning analysis (VPA) analysis in this study showed that high-level N deposition reduced the contribution rate of soil microorganisms to vegetation community variation in grazing sheep and mowing grassland. In contrast, the contribution rate of soil properties on the vegetation community increased significantly [21,87].

## 5. Conclusions

This study showed that different grassland utilization patterns and N deposition levels and their interactions significantly influenced *Leymus chinensis* grassland. Specifically, N deposition increased soil inorganic N and SOM, and decreased pH, while grazing sheep and mowing had the opposite effects. Grazing sheep and mowing showed significant interactions with N deposition, which suggests that N deposition could, to some extent, compensate for the adverse effects of grazing sheep and mowing on the soil.

Grazing sheep reduced the importance of *Poaceae* in the community, leading to more significant variability in community structure for other species. N deposition increased the proportion of *Leymus chinensis* in the community, decreased community diversity, enhanced the correlation between soil nutrients and community structure, and increased the contribution rate of soil environmental factors to changes in vegetation community structure. The mowing plots with N addition (N8) were associated with higher diversity and lower ACE than plots without any treatments (CK-N0).

In general, increasing N deposition in *Leymus chinensis* grassland used for grazing and mowing increased soil inorganic N and organic matter (SOM) content, and increased the amount and activity of microorganisms. To a certain extent, it also maintained community diversity and grassland productivity and contributed to the beneficial grazing or mowing and utilization of *Leymus chinensis* grassland.

**Author Contributions:** Conceptualization, D.H. and L.L.; methodology, C.Z.; software, C.Z.; validation, C.Z. and X.W.; formal analysis, S.Z. and D.H.; investigation, Q.L. and Y.J.; resources, S.Z. and L.L.; data curation, C.Z. and S.Z.; writing—original draft preparation, C.Z. All authors have read and agreed to the published version of the manuscript.

**Funding:** This research was supported by the earmarked fund for China Forage and Grass Research System (CARS-34) and the Climate-Smart Grassland Ecosystem Management Project (10006-P166853-2021-PIR-WB-China).

**Institutional Review Board Statement:** Not applicable.

**Data Availability Statement:** Not applicable.

**Conflicts of Interest:** The authors declare no conflict of interest.

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
