# Peer review of "Effects of Grazing Sheep and Mowing on Grassland Vegetation Community and Soil Microbial Activity under Different Levels of Nitrogen Deposition"

_agriculture, doi:10.3390/agriculture12081133_

Round 1
Reviewer 1 Report
The manuscript “Effects of grazing and mowing on grassland vegetation community and soil microbial activity under different levels of nitrogen deposition by Zhou et al. is a well-organized manuscript. However, some improvements need to be implemented not to confuse the readers.
The animal model used for mowing and grazing is very important to distinguish the effect of heavy livestock. Please add “sheep” along the manuscript (title, abstract, results, discussion) to emphasize that these results are related to their effect.
L75-80 Please add appropriate references to the site description data.
The authors need to provide a proper justification for the N levels applied.
The methods could be improved with a table or figure for the experimental design, specifying the sample size at each treatment. Especially because it is not a full factorial design.
Table 1. The ANOVA part (right side) is confused, especially because the table reflects two effects and the interaction is not very clear, and the “or” can be confusing.
L329, 349 – The science is not a belief; please rephrase to avoid this term.
The authors claim that sheep had fasted in captivity before grazing to minimize the effects of the sheep's feces and urine. However, there is no discussion of this potential bias. Along with the fact that these results are exclusive for these animals, with a specific weight, animal condition, and hoof shape.
The conclusions are very extensive, and some parts are more discussion-like (i.e., 432-433).
L438-443. This is a biased conclusion. The authors showed previous extended discussion regarding mowing and grazing intensity, and in the end, are very reductionist.
Author Response
Thank you for your valuable advice. Please see the attachment.

Reviewer 2 Report
The manuscript contains data on the simulation of the effect of nitrogen deposition on soil inorganic nitrogen content, pH, microbial activity and vegetation in grazed and mowed pastures.
It can be better after taking into account the following comments:
1. The Material and Methods chapter should be detailed.
2. Information should be provided on the number of sheep in G1 and G2 plots and on the grazing time.
3. Fertilization with increasing urea doses was used in the research, which is to simulate nitrogen deposition.
4. However, the accompanying fertilization with potassium, phosphorus and magnesium fertilizers was not specified. Nobody fertilizes permanent grassland with nitrogen fertilizers only.
5. In addition, there are many punctuation errors in this chapter. There should always be a space between the number / digit and the unit.
6. Subsection 2.3. it should be written in more detail.
7. Although the authors cite the methodical literature, a short synthetic description of the methods would be more advantageous for the manuscript.
8. Table 1 is illegible, numbers converge with standard deviation.
9. In Table 1, mass units are incorrectly spelled. Instead of Kg, there should be kg.
10. In the name of the first column of this table, soil should be changed to Soil.
11. All drawings are of poor quality.
12. In the Material and Method section, explain how the data for Figure 6 was obtained.
13. Chapter References should be supplemented by inserting doi for each item.
Author Response

(The authors gave the same response as above.)

Round 2
Reviewer 1 Report
The authors have included all suggestions.
Reviewer 2 Report
The current version of the article is correct. The manuscript may be published.